# Longitudinal analysis at three oral sites links oral microbiota to clinical outcomes in allogeneic hematopoietic stem-cell transplant

Vitor Heidrich,[1,2] Franciele H. Knebel,[1] Julia S. Bruno,[1] Vinícius C. de Molla,[3,4] Wanessa Miranda-Silva,[1] Paula F. Asprino,[1] Luciana Tucunduva,[5] Vanderson Rocha,[6,7] Yana Novis,[5] Eduardo R. Fregnani,[1] Celso Arrais-Rodrigues,[3,4] Anamaria A. Camargo[1]

**ABSTRACT**   Allogeneic hematopoietic stem-cell transplant (allo-HSCT) is potentially curative for several hematological disorders. Before stem-cell infusion, recipients undergo a conditioning regimen with chemo/radiotherapy and immunosuppressants, requiring the use of antibiotics to treat and prevent infections. This regimen promotes drastic alterations in the recipient's microbiota, including the oral microbiota, which have been associated with allo-HSCT complications and poor outcomes. However, long-term longitudinal studies on the oral microbiota of allo-HSCT recipients are scarce and disregard the existence of distinct microbiota within the oral cavity. Here, we used 16S rRNA gene sequencing to characterize the longitudinal microbiota dynamics in a prospective cohort of 31 allo-HSCT recipients at three oral sites (gingival crevicular fluid, oral mucosa, and supragingival biofilm). We found declines in bacterial diversity and major shifts in microbiota composition in all oral sites during allo-HSCT, including blooms of potentially pathogenic genera. These blooms in some cases preceded respiratory infections caused by the blooming bacteria. We also noticed that differences in microbiota composition between oral sites were lost during allo-HSCT. Overall, after allo-HSCT, the distinct oral microbiota returned to their preconditioning state, although at variable levels of competence per patient. After stratifying patients based on recovery levels, we found that recoverers of the oral mucosa microbiota composition had earlier reconstitution of normal blood leukocyte counts. Most notably, oral mucosa microbiota recovery was an independent biomarker of better allo-HSCT outcomes. This study highlights the potential clinical impact of the oral microbiota in the allo-HSCT setting and the clinical value of tracking oral microbiota changes during allo-HSCT.

**IMPORTANCE**   The oral cavity is the ultimate doorway for microbes entering the human body. We analyzed oral microbiota dynamics in allogeneic hematopoietic stem-cell transplant recipients and showed that microbiota injury and recovery patterns were highly informative on transplant complications and outcomes. Our results highlight the importance of tracking the recipient's microbiota changes during allogeneic hematopoietic stem-cell transplant to improve our understanding of its biology, safety, and efficacy.

**KEYWORDS**   oral microbiome, 16S rRNA gene sequencing, allogeneic hematopoietic cell transplant, microbiome stability, blooming of bacteria, biomarkers, clinical outcomes

Countless microbes from food, air, and our physical/biological environment arrive in our mouths daily. However, only a small subset of these microbes can colonize the oral cavity to compose the oral microbiota (1). This constant contact with non-resident

Address correspondence to Anamaria A. Camargo, anamaria.acamargo@hsl.org.br.

The authors declare no conflict of interest.

See the funding table on p. 19.

microbes and frequent exposure to other insults (e.g., toothbrushing) made the human oral microbiota remarkably stable and resilient to external perturbations (2).

Residing oral microbes organize in biofilms, creating oxygen gradients that allow colonization by both anaerobic and aerobic bacteria (1). Differences in moisture, topography, and tissue type (shedding vs non-shedding), among others, make each oral site home to distinct bacterial communities (1, 3) with main compositional differences existing between mucosa-associated and teeth-associated microbiota (4).

These distinct oral microbiota are important regulators of human health, as they have been associated with different local and systemic disorders (5). While the supragingival biofilm (SB) is causally linked to the pathogenesis of dental caries (6), bacteria at the gingival crevice, an oxygen-limited environment bathed in immune exudate (gingival crevicular fluid, GCF), are linked to periodontitis (7) and may cause bacteremia by translocation to the circulation across the thin gingival crevice epithelium (8). Oral bacteria can further facilitate systemic reach by producing molecules that increase vascular permeability (5). Using this strategy, oral *Porphyromonas gingivalis* is able to colonize the brain, contributing to the pathogenesis of Alzheimer's disease (9).

Allogeneic hematopoietic stem-cell transplant (allo-HSCT) is used to treat malignant (e.g., acute myeloid leukemia) and non-malignant (e.g., aplastic anemia) hematological disorders (10). The goal of allo-HSCT is to eradicate malignant/defective cells and to replace an abnormal hematopoietic and immune system (11). Allo-HSCT recipients undergo a conditioning regimen with chemo/radiotherapy that reduces disease burden and provides sufficient immunoablation to allow donor stem-cell engraftment (12). After engraftment, the graft-vs-tumor/autoimmunity effect further promotes disease eradication, and the hematopoietic/immune function gradually reconstitutes (13). Besides chemo/radiotherapy, allo-HSCT recipients are treated with immunosuppressants to prevent engraftment failure and graft-vs-host disease, and antibiotics to prevent and treat opportunistic infections during immunosuppression (13, 14).

Allo-HSCT is considered one of the most severe perturbations the immune system undergoes in the therapeutic setting (15). Since the immune system regulates microbiota composition (16) and chemotherapy (17), radiotherapy (18), and antibiotics (19) have detrimental effects on the microbiota, drastic alterations in the gut microbiota have been reported in allo-HSCT recipients, including loss of bacterial diversity and blooms of potentially pathogenic species (20). Recent evidence shows these alterations extend to other microbiota (21), including the relatively more stable oral microbiota (22–26). More importantly, the pre-transplant microbiota and the extent of microbiota damage during allo-HSCT are associated with allo-HSCT complications and outcomes, so that gut and oral microbiota provide biomarkers in the allo-HSCT setting (24, 25, 27–30).

The stability of the oral microbiota (5) and its associations with allo-HSCT outcomes offer a unique opportunity to identify predictive biomarkers and develop therapeutic interventions to promote oral health in allo-HSCT recipients, potentially improving allo-HSCT safety and efficacy. However, so far, oral microbiota studies in allo-HSCT recipients evaluated single oral sites, not leveraging the ease of sampling of different oral compartments (22–26, 30). In addition, although a causal link between post-transplant gut microbiota recovery and improved clinical responses to allo-HSCT has been suggested (15), oral microbiota recovery trajectories after allo-HSCT were not thoroughly characterized and their association with allo-HSCT outcomes remains unknown.

To obtain a more in-depth understanding of oral microbiota dynamics during and after allo-HSCT and to test whether oral microbiota recovery is associated with allo-HSCT outcomes, we profiled the oral microbiota of a Brazilian cohort of allo-HSCT recipients. We collected over 440 samples encompassing five timepoints and three oral sites: gingival crevicular fluid, oral mucosa (OM), and supragingival biofilm, which allowed a longitudinal anatomically aware analysis of the oral microbiota. We used 16S rRNA gene sequencing to characterize diversity, compositional, and taxonomical changes in oral microbiota during allo-HSCT and after engraftment. We associated these changes with

antibiotic usage and allo-HSCT complications. Finally, we evaluated recovery trajectories after allo-HSCT to associate oral microbiota recovery with allo-HSCT outcomes.

## MATERIALS AND METHODS

### Patients' clinical characteristics

In this prospective cohort study, 31 patients undergoing allo-HSCT at Hospital Sírio-Libanês (São Paulo, Brazil) were recruited between January 2016 and April 2018. The median age was 50 years, most patients were male (55%), and acute leukemia was the most common underlying disease (58%; 11 acute myeloid leukemia and 7 acute lymphocytic leukemia cases). Most patients underwent reduced intensity conditioning (61%) and received grafts from peripheral blood (68%). Patient clinical information is summarized in Table S1. The follow-up cutoff date was 09 May 2021.

### Antibiotic usage analysis

Antibiotic prescriptions were retrieved retrospectively from clinical records. Information spanning 30 days before preconditioning sampling and 100 days after the stem-cell infusion was collected to build timelines of antibiotic usage for each patient (Timelines of antibiotic usage in File S1). A ridgeline plot of antibiotic usage detailing all antibiotics and antibiotic classes used showed antibiotics prescription concentrates in the few weeks immediately after infusion (Fig. S1), with only 5/31 patients receiving antibiotics before preconditioning (File S1). Due to the sparse use of antibiotics before preconditioning and the unlikely effect of antibiotics received months after allo-HSCT on clinical outcomes, antibiotic usage was analyzed considering only the time window between preconditioning and 30 days after engraftment (a patient deceased during this period was excluded from the analysis). For each patient, the length of days under antibiotic therapy (length of therapy, LOT) and the number of agent days under antibiotic therapy (days of therapy, DOT) were calculated, as defined previously (31). To evaluate the impact of specific antibiotic classes on microbiota dynamics, patients were further classified according to antibiotic class usage during the period of interest. Only antibiotic classes received by at least 20% of our patients (6/30) were considered in this analysis. In addition to individual antibiotic prescriptions, all patients underwent standard antimicrobial prophylaxis with antibiotic, antiviral, and antifungal drugs. Because the standard antibiotic prophylaxis protocol in our institution comprises oral levofloxacin and sulfamethoxazole-trimethoprim, their use was not considered in the antibiotic usage analysis.

### Sample collection

Patients were examined frequently by an oral medicine specialist throughout the hospitalization period. The standard oral hygiene protocol comprised toothbrushing with fluoridated toothpaste and 0.12% chlorhexidine mouthwash. Samples were collected at least 6 hours after the last oral hygiene procedure by an oral medicine specialist at three oral sites. GCF samples were collected by inserting absorbent paper points in the gingival crevice; OM samples were collected by swabbing bilateral buccal mucosa, alveolar mucosa of the jaws, and tongue dorsum; SB samples were collected by swabbing all vestibular enamel surface. Samples were dry-stored at −20°C.

### DNA extraction and 16S rRNA gene amplicon sequencing

Samples were brought to room temperature. Bacterial cells were recovered from swabs or paper points by vortexing in 600 or 800 µL TE buffer (10 mM Tris; 1 mM EDTA; pH 8.0), respectively. Samples were transferred to a new tube, supplemented with 6 µL (OM and SB) or 8 µL (GCF) PureLink RNase A (20 mg/mL; Invitrogen), and DNA was extracted using the QIAamp DNA Mini Blood kit (Qiagen) following the manufacturer's protocol (Buccal Swab Spin Protocol).

Bacterial communities were profiled by 16S rRNA gene amplicon-sequencing as described in detail previously (32). In short, amplicon libraries were prepared with 12.5 ng of total DNA and pre-validated V3V4 primers (33) following Illumina's protocol (preparing 16S ribosomal RNA gene amplicons for the Illumina MiSeq System). Amplicons were sequenced on the Illumina MiSeq platform using the MiSeq Reagent Kit v3 (600 cycle) (Illumina). All laboratory work was done in the Center of Molecular Oncology (Hospital Sírio-Libanês, São Paulo, Brazil)

## Bioinformatics pipeline

Reads were demultiplexed using the MiSeq Reporter Software. Primers were removed and low-quality 3′ ends were trimmed using seqtk (34). Next, reads were processed using QIIME 2 (v2019.10.0) as schematized in Fig. S2a (35). In detail, amplicon sequence variants (ASVs) were generated using the DADA2 pipeline (via q2-dada2), which includes the removal of low-quality reads, denoising, merging, and removal of bimeras (36). Chimeric ASVs were further filtered out using a reference-based approach with VSEARCH (37) (via q2-vsearch) and SILVA database (v132) (38). Taxonomic assignment of ASVs was also performed with VSEARCH (37) (via q2-feature-classifier) and SILVA (v132) (38). Finally, non-bacterial ASVs were removed (via q2-feature-table). QIIME 2 outputs were transferred to the R environment (39) using the qiime2R R package (40) and analyzed for microbiota profiling with custom R scripts as detailed below.

## Microbiota and statistical analyses

The total number of reads of the sample with the lowest number of reads (3,578 reads) among the samples included in the microbiota profiling analyses was used as $C_{min}$ for scaling with ranked subsampling (SRS) normalization prior to diversity analyses (41). Diversity was measured by the Gini-Simpson index (42) using the phyloseq R package (43). Longitudinal diversity variations were evaluated by calculating diversity resistance, resilience, and stability (44, 45) (see Methods in File S3). The compositional dissimilarity between samples was measured by the weighted UniFrac distance (46) using the rbiom R package (47). Longitudinal compositional variations were evaluated by calculating compositional stability (see File S3). Multiple linear regression was used to evaluate whether antibiotic usage was associated with diversity stability and compositional stability (see File S3). Recovery to baseline composition was defined as the distance between samples collected at preconditioning and 30 days after engraftment <0.5.

Taxonomic nomenclature was homogenized prior to all taxonomic analyses (see File S3). Taxa relative abundance (RA) plots included only the most relevant genera according to the criteria specified in figure legends. Differential abundance analysis was performed using Analysis of Compositions of Microbiomes with Bias Correction (ANCOM-BC) (48) with genera present in ≥25% of the samples being compared. Genera abundance differences between groups at $q$-value < 0.05 (Bonferroni correction) were considered statistically significant, including ANCOM-BC structural zeroes.

Associations between oral microbiota composition recovery or clinical parameters with allo-HSCT outcomes were determined using univariate Cox proportional-hazards regression (49) or univariate Fine-Gray competing risk regression (50). Cox models were used to evaluate overall survival (OS) and progression-free survival (PFS), while Fine-Gray models were used to evaluate the risk of transplant-related death (with relapse mortality as competing risk) and the risk of underlying disease relapse (with transplant-related mortality as competing risk). Multivariate analysis was used to evaluate oral microbiota composition recovery and correct for clinical parameters significantly associated with the outcome ($P$-value < 0.05) in the univariate analysis. Patients experiencing the event before oral microbiota composition recovery evaluation were excluded from univariate and multivariate analyses. The Strengthening The Organizing and Reporting of Microbiome Studies(STORMS) checklist for this work is available at https://github.com/vitorheidrich/oral-microbiota-hsct.

## RESULTS

### Samples collected and sequencing output

We collected samples from three oral sites (GCF, OM, and SB) at five timepoints during allo-HSCT: preconditioning (P), aplasia (A), engraftment (E), 30 days after engraftment (E30), and 75 days after engraftment (E75). Since most patients were discharged shortly after engraftment, the exact date of sample collection varied for E30 (20–45 days after engraftment) and E75 (60–131 days after engraftment) samples, as indicated in Fig. S3. Premature death after allo-HSCT hampered the collection of the E30 sample for patient #3 and E75 samples for patients #1, #2, #3, #21, and #31 (Fig. S3). In addition, the E75 sample from patient #9 was excluded due to low DNA yield. Overall, 444 samples were successfully processed and sequenced for microbiota profiling.

We generated a total of 53,253,725 V3V4 16S rRNA reads (median per sample: 104,230.5; range: 2,059–502,409; Fig. S2b). After filtering, 31,343,619 reads (59%; Fig. S2c and d) were retained (median per sample: 63,075.5; range: 87–310,082; Fig. S2e), corresponding to 4,046 ASVs. Using SRS curves (51) (Fig. S4), we established a minimum sequencing depth cutoff of 3,000 reads, and four low-depth samples were excluded from further analysis (patient #1, OM, P; #5, OM, E; #6, OM, E; #25, SB, E). We proceeded to profile the oral microbiota during allo-HSCT with 440 samples.

### Compositional differences between distinct oral microbiota during allo-HSCT and after engraftment

We first assessed microbiota compositional differences between oral sites at each allo-HSCT timepoint. Visually, all three oral microbiota occupied a similar compositional space throughout allo-HSCT (Fig. 1a). Nevertheless, similarly to what is observed in healthy adults (4), each oral site contained a significantly different microbiota composition at P (PERMANOVA, GCF vs OM: $P$-value = 0.001; GCF vs SB: $P$-value = 0.002; OM vs SB: $P$-value = 0.018). Noteworthy, these differences progressively diminished in subsequent timepoints until E30 and were partially recovered at E75 (Fig. 1b). Calculation of the minimum compositional distance between oral sites for each patient confirmed lower compositional distance between sites after P (Fig. 1c).

Differential abundance analysis at the genus level using ANCOM-BC revealed a similar picture (Fig. 1d). As expected, all three oral microbiota showed many distinguishing genera at P. For example, we observed a higher abundance of *Actinomyces* in the SB as compared to GCF and a higher abundance of *Solobacterium* in the OM as compared to SB (Fig. S5). *Actinomyces* spp. are early colonizers of the SB with a crucial role in ecological succession during SB maturation (52). On the other hand, *Solobacterium moorei*, the only known species in the *Solobacterium* genus, is a halitosis-associated bacteria typically found in the tongue dorsum (53), a site contemplated in OM samples. However, a smaller number of differentially abundant genera was observed in subsequent timepoints, with a slight increase in the number of differentially abundant genera between sites at E75, illustrated by the reappearance of *Solobacterium* as an OM-associated genus (Fig. S5).

### Oral microbiota dynamics during allo-HSCT and after engraftment

We next characterized microbiota diversity dynamics at each oral site during allo-HSCT and after engraftment. As previously shown for OM (25) and SB (24), GCF presented a stepped decline in diversity up to E (Fig. 2a). By extending this analysis to the post-engraftment period for all oral sites, we observed a gradual recovery of diversity, with baseline levels almost fully reestablished around E75.

We then applied key concepts from ecology (45) for a more in-depth characterization of diversity dynamics during allo-HSCT. By considering allo-HSCT as a perturbation relieved immediately after engraftment, we calculated diversity resistance (inversely proportional to the diversity loss up to E), resilience (rate of diversity gain after E), and stability (combined effect of resistance and resilience) to allo-HSCT for each patient (Fig. S6a; see File S3). GCF showed higher diversity resistance than OM and SB (Fig. 2b), in line

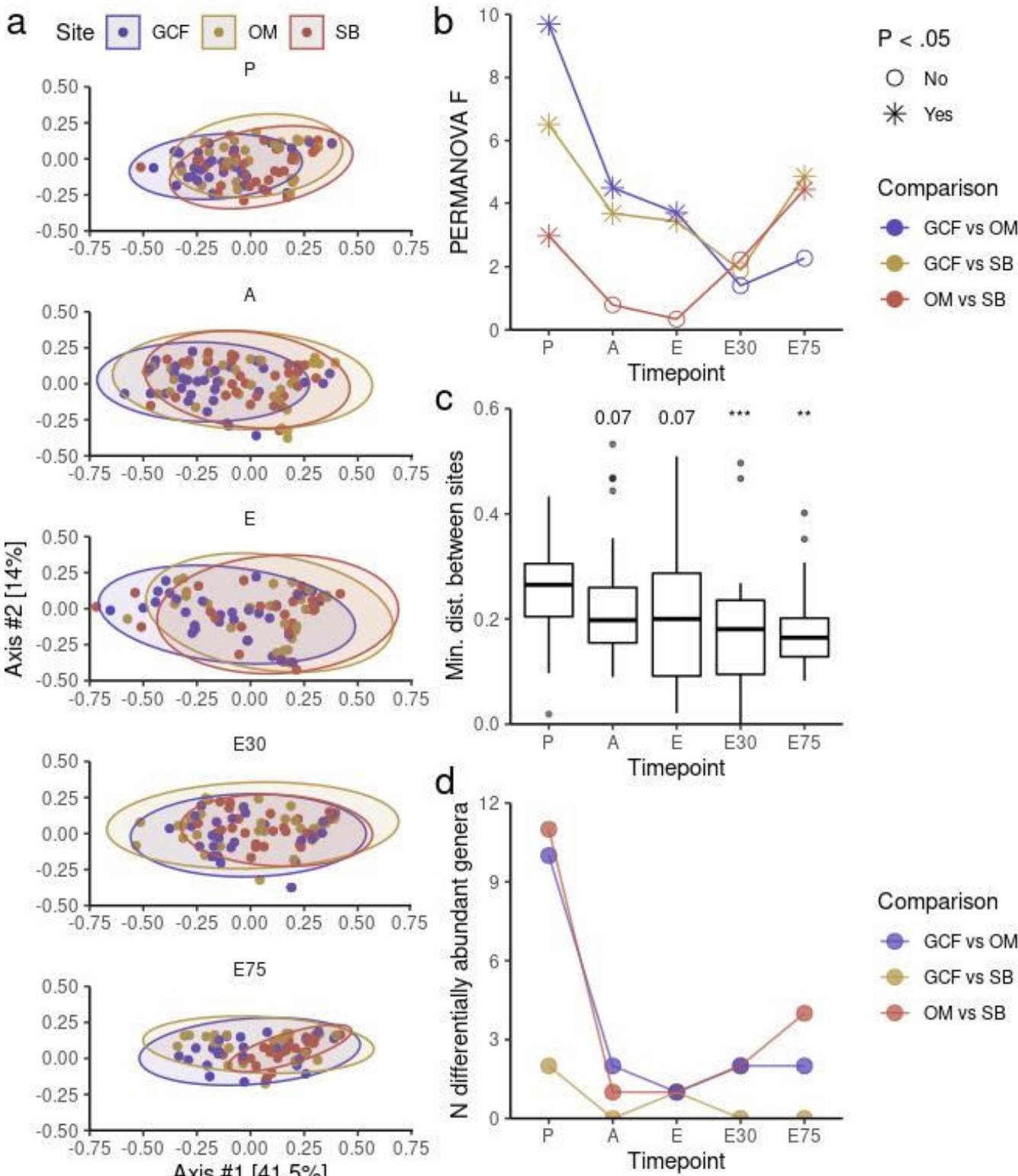

**FIG 1** (a) Principal Coordinate Analysis (PCoA) of microbiota distances (weighted UniFrac) between oral sites for each timepoint. Ellipsoids indicate 95% confidence intervals. (b) Magnitude (PERMANOVA F) of distances (weighted UniFrac) between oral sites per timepoint. (c) Minimum distance (weighted UniFrac) between oral sites within patients per timepoint. Mann-Whitney *U* test was used with preconditioning (P) as the reference for comparisons. (d) Number of differentially abundant genera (ANCOM-BC) between oral sites per timepoint. E30, 30 days after engraftment and E75, 75 days after engraftment. **\**P*-value < 0.01 and \*\*\**P*-value < 0.001.

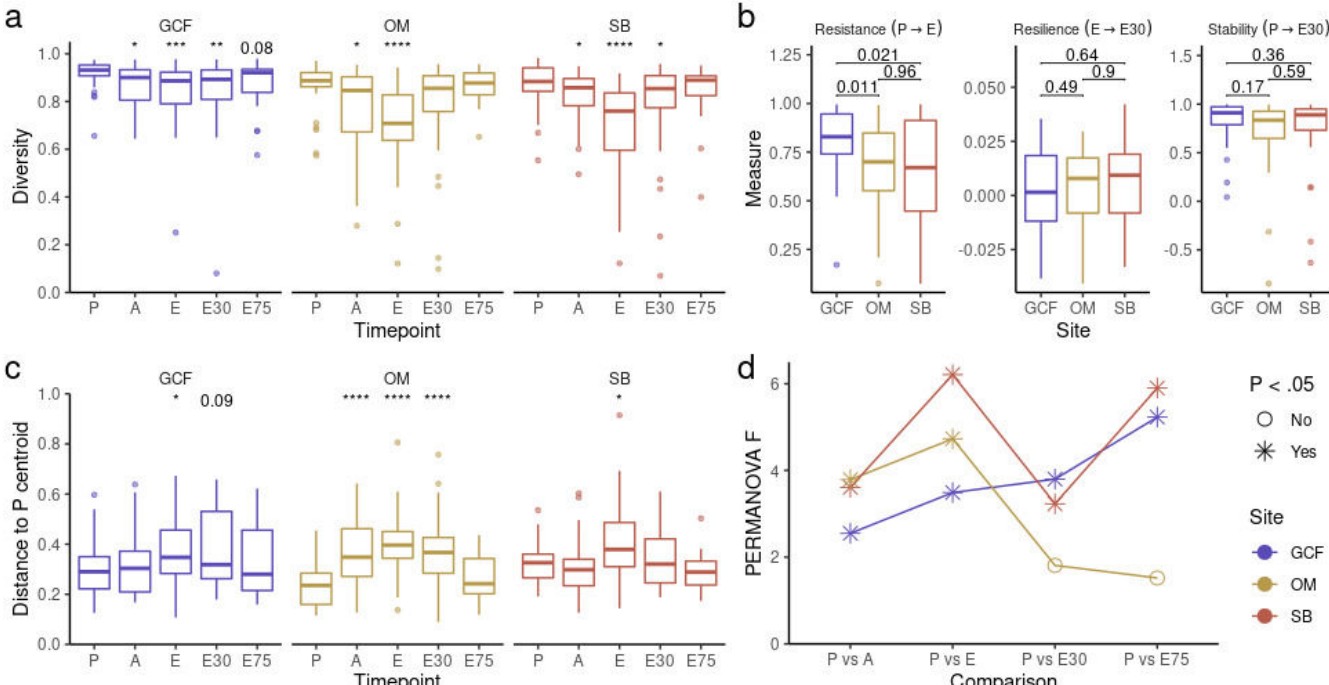

**FIG 2** (a) Diversity (Gini-Simpson) per timepoint for each oral site. Mann-Whitney *U* test was used with preconditioning (P) as the reference for comparisons. (b) Diversity resistance, resilience, and stability (see Materials and Methods) per oral site. Mann-Whitney *U* test was used. Distance to P centroid (weighted UniFrac) per timepoint for each oral site. Mann-Whitney *U* test was used with P as the reference for comparisons. (d) Magnitude (PERMANOVA F) of distances (weighted UniFrac) between P and other timepoints for each site. E30, 30 days after engraftment and E75, 75 days after engraftment. *P-value < 0.05; **P-value < 0.01; ***P-value < 0.001; and ****P-value < 0.0001.

with the less pronounced loss of diversity observed in this oral site at E (Fig. S6b). All oral sites presented equivalent levels of diversity resilience and stability (Fig. 2b), in line with the similar levels of diversity after engraftment observed for all oral sites (Fig. S6b).

Next, we characterized compositional changes in each oral site during allo-HSCT and after engraftment. The compositional distance to P centroid increased up to E and decreased in the post-engraftment period, indicating a displacement from and posterior recovery to baseline compositions (Fig. 2c). However, when comparing the compositional distance from P to all other timepoints using PERMANOVA tests, we observed that GCF and SB post-engraftment samples still showed significantly different compositions after engraftment compared to P, while OM samples more fully recovered their preconditioning state (Fig. 2d). Finally, in analogy to diversity stability, we calculated the compositional stability for each patient (see File S3). As observed for diversity stability, all oral sites showed equivalent levels of compositional stability (Fig. S6c).

## Oral taxa abundances during allo-HSCT and after engraftment

As expected, all oral sites presented a high relative abundance of commensal bacteria at P (Fig. 3a; Fig. S7). For instance, *Veillonella* and *Streptococcus*, genera with high relative abundance in all oral sites of healthy adults (4), occupied either the first or second position in terms of mean relative abundance at P in all three oral sites (Fig. 3b). However, there were several changes in the ranking of the most abundant taxa (on average) across timepoints (Fig. 3b; Fig. S7), pointing to drastic taxonomic composition changes during allo-HSCT. There are some noteworthy examples, such as *Streptococcus* in SB, which went from first in the relative abundance ranking at P to the 11th position at E. Interestingly, *Streptococcus* recovered its initial ranking position after engraftment (first position at E30 and E75). On the other hand, some genera were close to absent in P and only emerged in the subsequent timepoints. For instance, *Enterococcus* and *Lactobacillus*, both potentially

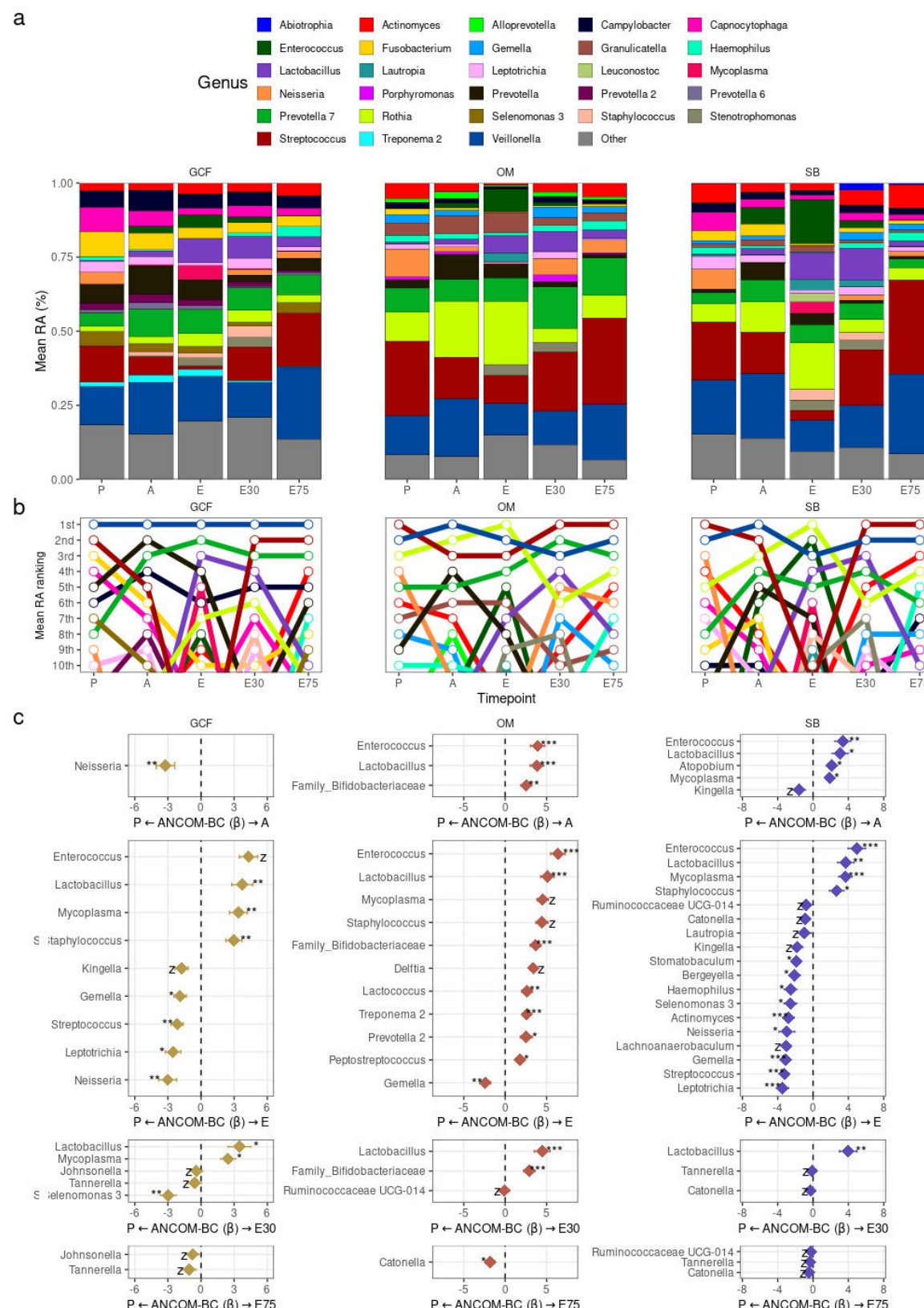

**FIG 3** (a) Mean genera RA per timepoint for each oral site. Genera with >2% mean RA in any combination of oral site and timepoint are shown. (b) Mean genera RA ranking per timepoint for each oral site. Top-10 genera are shown. (c) Differentially abundant genera (ANCOM-BC) between P and other timepoints for each site. E30, 30 days after engraftment and E75, 75 days after engraftment. *$q$-value < 0.05; **$q$-value < 0.01; ***$q$-value < 0.001; and z, ANCOM-BC structural zero.

pathogenic genera in the oral microbiota (54, 55), showed low mean relative abundance at P but were among the most abundant genera in all sites at E.

Differential abundance analysis at the genus level using ANCOM-BC with P as reference for comparisons confirmed these results and showed several additional differentially abundant genera (Fig. 3c). The number of differentially abundant genera at each timepoint was consistent with the compositional displacement and recovery aforementioned, with a maximum of differentially abundant genera at E (Fig. S8). Although there were considerably fewer differentially abundant genera after engraftment, some differences persisted. For instance, we observed a decreased abundance of *Catonella* in OM and SB, and of Tannerella in GCF at E75, suggesting a long-lasting reduction of these genera caused by allo-HSCT.

## Emergence of opportunistic genera and allo-HSCT complications

The emergence of opportunistic genera during allo-HSCT can be more rigorously quantified by assessing taxa blooms, defined as a taxon relative abundance increase from <1% at P to dominance levels (≥30%) at any subsequent timepoint. We have previously shown, by analyzing this same cohort, blooms of specific genera occurring in SB during A and E (24). We now extended this analysis to other oral sites and to the post-engraftment period. Overall, we detected 81 blooms, involving 22 genera and 27/31 patients. All oral sites showed several blooming events, but SB blooms were more frequent (SB: $n = 35$; GCF: $n = 24$; OM: $n = 22$; Fig. 4a) and significantly more prevalent (SB: 23/31; GCF: 14/31; MO: 16/30; chi-square test, *P*-value = 0.022). Blooms typically occurred at E (53% of events; Fig. 4b) and were rapidly resolved in the post-engraftment period.

*Lactobacillus* (15%), *Enterococcus* (12%), and *Staphylococcus* (10%) were the genera most frequently observed in blooming events in the oral microbiota during allo-HSCT (Fig. 4c). But oral sites differed in the genera typically associated with blooms (Fig. 4d). SB showed mainly *Enterococcus* (seven events) or *Lactobacillus* (6) blooms, while GCF showed mostly *Staphylococcus* (4) or *Lactobacillus* (4) blooms. In contrast, OM blooms showed a less clear signal of blooming genera. Nevertheless, some patients presented concomitant blooms of the same genus in all oral sites.

We noticed that many of the blooming genera are potentially pathogenic for allo-HSCT recipients. For instance, *Staphylococcus* genus contains species related to several infections, including hospital-acquired pneumonia (56), an allo-HSCT complication with 15%–30% incidence (57). Therefore, we evaluated whether blooming events in the oral microbiota were associated with respiratory infections in our cohort. Between P and E75, only 3/31 patients presented bacterial respiratory infections (patients #1, #2, and #7). All three patients showed blooms of genera in the oral microbiota during allo-HSCT. Specifically, patient #1 presented blooms of *Enterococcus* (in GCF and SB at E) and Acetobacter (in GCF and SB at E30), patient #2 presented blooms of *Stenotrophomonas* (in all oral sites at E) and *Mycoplasma* (in GCF at E), and patient #7 presented blooms of *Mycoplasma* (in OM and SB at E). Interestingly, patients #1 and #2 presented blooms of the same genus identified in the microbiological exam of their respiratory tract samples: *Enterococcus* and *Stenotrophomonas*, respectively. Importantly, these blooms preceded the clinical manifestation of the respiratory infection by 1 and 2 weeks, respectively, suggesting a potential oral origin for the bacteria associated with the respiratory infections in these cases. On the other hand, patient #7 developed a respiratory infection caused by *Escherichia coli* between E30 and E75, which was unrelated to the blooms detected for this patient.

Given the apparent translocation of abundant oral bacteria to the respiratory tract in our cohort and the well-known association between intestinal dominance and bacteremia during allo-HSCT (58), we also tested whether blooming events in the oral microbiota were associated with bacteremia events. Positive blood cultures for bacteria were detected for 15/31 patients between P and E75. We did not find an association between oral microbiota blooms and altered odds of bacteremia (Fisher's exact test, GCF bloom: OR = 3.17, *P*-value = 0.156; OM bloom: OR = 2.25, *P*-value = 0.299; SB bloom: OR =

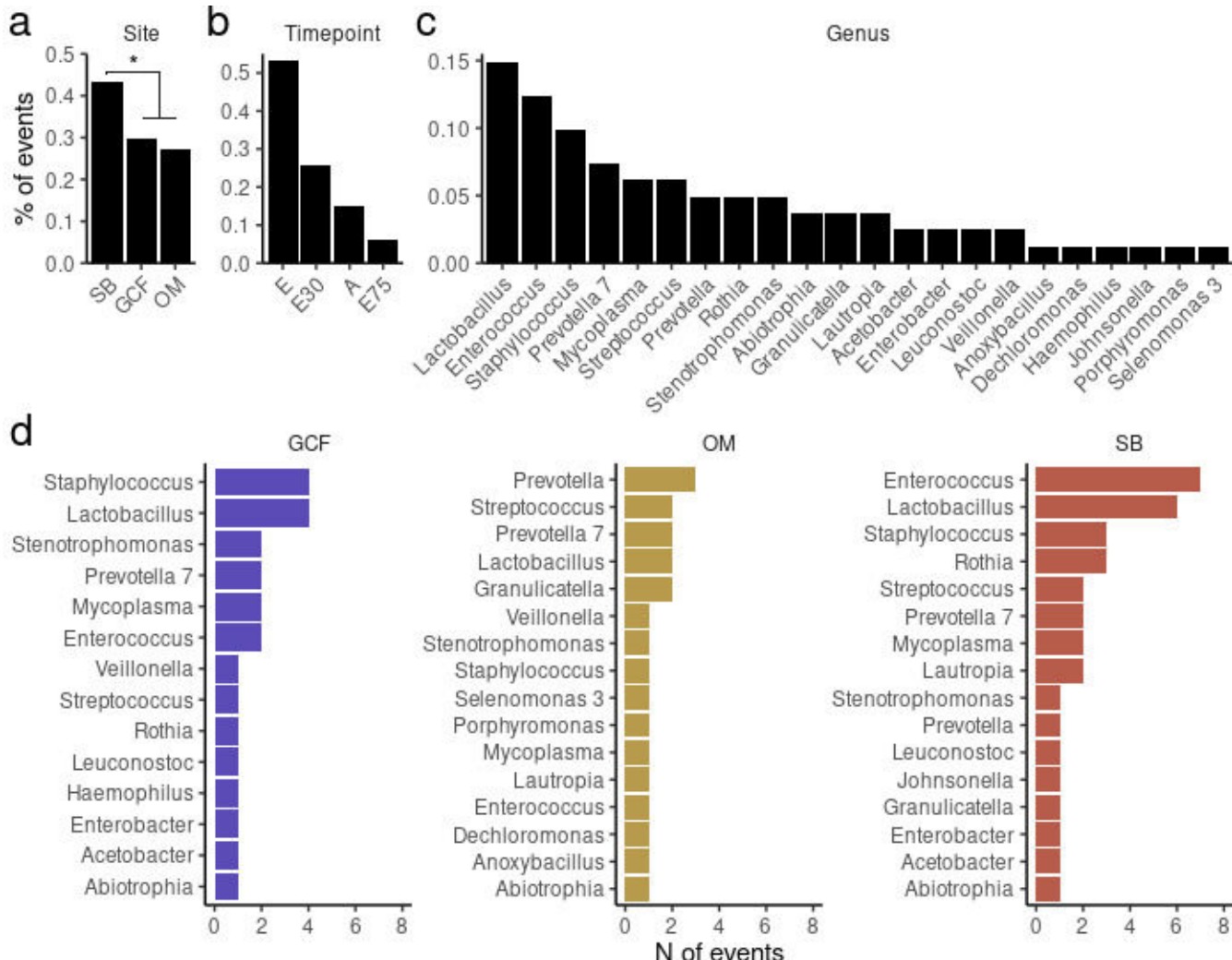

**FIG 4** (a–c) Proportion of blooming events per oral site (a), timepoint (b), and genus (c). (d) Number of blooming events per genus in each oral site. E30, 30 days after engraftment and E75, 75 days after engraftment.

0.92, *P*-value = 1; any site bloom: OR = 3.12, *P*-value = 0.600). We detected a single case in which the blooming of a genus in the oral microbiota preceded a bacteremia event with the same genus involved. In detail, patient #14 presented blooms of *Enterococcus* in GCF and SB at A, which preceded positive blood cultures for *Enterococcus* by 1.5 weeks.

## Impact of antibiotic usage on oral microbiota dynamics

To investigate the impact of antibiotic usage on oral microbiota dynamics and blooming events during allo-HSCT, we analyzed antibiotic usage data between P and E30 (see Materials and Methods). Antibiotic usage varied widely across patients in terms of LOT (range: 0–58 days; median: 15.5 days) and DOT (range: 0–112 days; median: 22 days) (Table S1). Overall, 17 antibiotic agents (range: 0–10; median: 3), spanning 12 antibiotic classes (range: 0–9; median: 3) were administered to our patients. The antibiotics administered to each patient are illustrated in Fig. 5a. Most patients received cefepime (73%) and meropenem (63%), making cephalosporins and carbapenems the most frequently used antibiotic classes: 73% and 63%, respectively (Fig. S9a). Glycopeptides and penicillins were also used in a considerable proportion of patients: 60% and 23%, respectively. All other antibiotic classes were used by less than 17% of our patients (Fig. S9b).

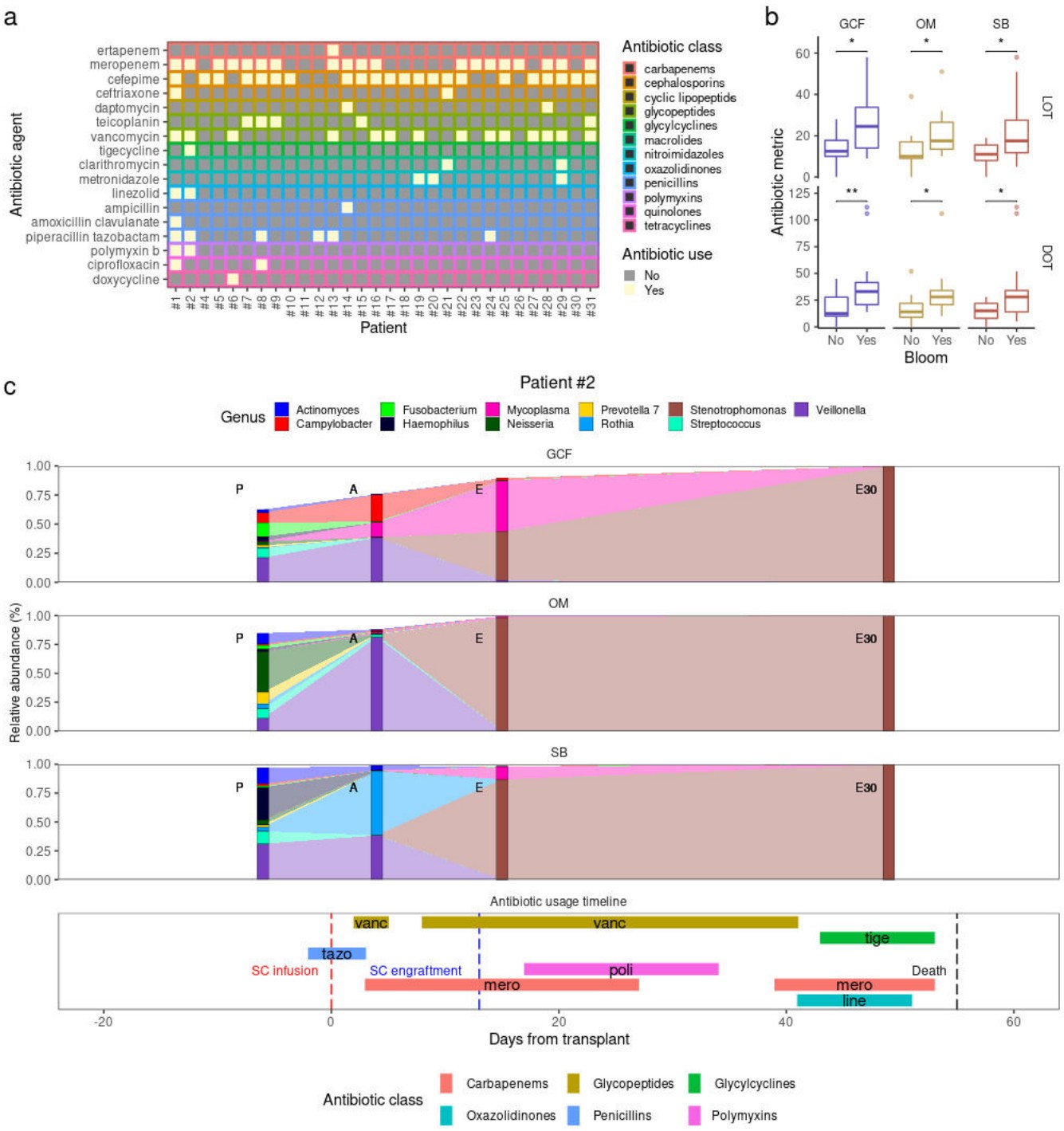

**FIG 5** (a) Antibiotic agents used by each patient between preconditioning (P) and 30 days after engraftment (E30). (b) Time of antibiotic administration (length of therapy and days of therapy) among patients showing and not showing blooms between P and E30. Mann-Whitney *U* test was used. (c) Patient #2: genera relative abundance dynamics for each oral site (top) and antibiotic usage timeline (bottom). Genera with >1% mean relative abundance in any combination of oral site and timepoint are shown. E75, 75 days after engraftment; SC, stem-cell; vanc, vancomycin; tige, tigecycline; tazo, piperacillin tazobactam; poli, polymyxin B; mero, meropenem; and line, linezolid.

First, to assess the effect of antibiotic usage in microbiota dynamics, we modeled diversity stability (which incorporates diversity resistance and resilience) and compositional stability using antibiotic usage information (Table S2). We found that DOT significantly predicted diversity stability during allo-HSCT for all oral sites, with

prolonged use of antibiotic therapy associated with lower diversity stability. However, the use of specific antibiotic classes was not associated with altered diversity stability (Table S2). On the other hand, DOT was not a predictor of compositional stability, but glycopeptide usage was significantly associated with decreased SB compositional stability (Table S2). In addition, we found non-significant associations at $P$-value < 0.1 between other antibiotic classes and decreased compositional stability in GCF (cephalosporins and penicillins) and SB (cephalosporins), while OM compositional stability was clearly less impacted by antibiotic usage during allo-HSCT (Table S2).

We next tested whether blooms at different oral sites were associated with antibiotic usage. E75 blooms were not considered in this analysis since our antibiotic usage survey focused on the period between P and E30 (see Materials and Methods). With one exception (glycopeptides and GCF blooms), the use of specific antibiotic classes was not associated with blooms, but patients experiencing blooms showed higher LOT and DOT (Fig. 5b), although it is not clear whether a more extended period under antibiotic therapy was the cause or consequence of the blooms.

GCF blooms were significantly associated not only with LOT and DOT but also with the use of glycopeptides [Fisher's exact test, odds ratio (OR) = 15.65, $P$-value = 0.006, $P$-adjusted = 0.025], which enabled the investigation of the relation between the timing of glycopeptide usage and GCF blooming events. GCF blooms occurred in 12 patients up to E30, out of which 11 used glycopeptides (vancomycin and/or teicoplanin) between P and E30. Notably, 10/11 patients who used glycopeptides and experienced GCF blooms received glycopeptides a few days before or during the interval in which the bloom was detected, indicating that glycopeptide usage during allo-HSCT may cause blooms of genera in the oral microbiota.

The relationship between glycopeptide usage and blooming events and its consequences can be illustrated by the genera composition trajectories and antibiotic usage timeline of patients #1 and #2. Patient #2 experienced *Stenotrophomonas* blooms in all sites at E, which occurred during the administration of vancomycin (Fig. 5c). Two weeks after these blooms, patient #2 developed a respiratory infection caused by *Stenotrophomonas maltophilia*, detected in microbiological exams of respiratory tract samples (e.g., bronchoalveolar lavage). Despite the intensification in the use of antibiotics, *Stenotrophomonas* levels only rose in the oral microbiota after E, reaching staggering levels at E30 (>95% relative abundance in all oral sites). Analysis at the ASV level revealed that *Stenotrophomonas* ASVs were absent in patient #2 at P (relative abundance = 0% in all oral sites). At A, during the first course of vancomycin (Fig. 5c), a *Stenotrophomonas maltophilia* ASV emerged in the SB (relative abundance = 0.02%). This ASV would later be responsible for the blooms at E and the domination observed at E30. Taken together, it is possible to speculate that the use of vancomycin during allo-HSCT might have allowed the emergence and the bloom of pathogenic *Stenotrophomonas maltophilia* in oral microbiota, which later translocated to the respiratory tract, causing a respiratory infection. Patient #1 presented a similar picture (Fig. S10), with the use of vancomycin followed by *Enterococcus* blooms and a subsequent respiratory infection caused by *Enterococcus faecium*. Notably, patients #1 and #2 died before E75, with death causes at least partially associated with their respiratory infections.

## Inter-patient variability in oral microbiota dynamics during allo-HSCT and after engraftment

To investigate inter-patient variability in oral microbiota dynamics during allo-HSCT and after engraftment, we assessed longitudinal changes in oral microbiota in a patient-centered analysis. Although most patients presented high diversity stability, which was achieved either by having high resistance, high resilience, or a balance between the two, some patients presented low diversity stability and even negative resilience values (Fig. 6a), indicating loss of diversity after E. Curiously, this inter-patient variability was not due to different levels of baseline diversity, since diversity at P was not correlated with

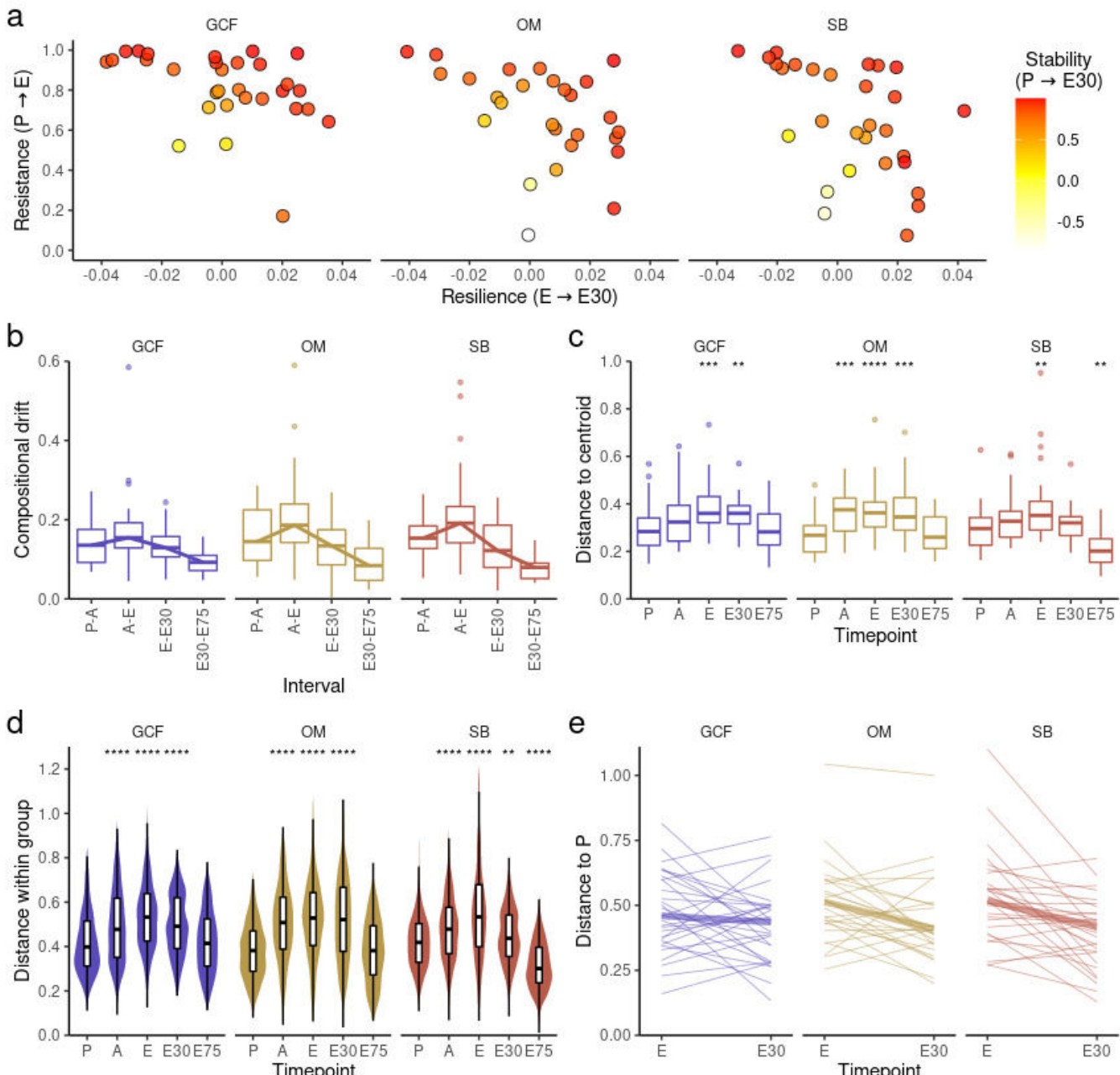

**FIG 6** (a) Relationship between diversity resistance, resilience, and stability values calculated for each patient. (b) Extent of compositional shifts (weighted UniFrac) between consecutive timepoints (adjusted for the time in days between timepoints) for each oral site. The line indicates the median value per interval. (c) Distance (weighted UniFrac) to timepoint centroid per timepoint for each oral site. Mann-Whitney $U$ test was used with preconditioning (P) as the reference for comparisons. (d) Pairwise distances (weighted UniFrac) per timepoint (all-against-all) for each oral site. Mann-Whitney $U$ test was used with P as the reference for comparisons. (e) Distance to P (weighted UniFrac) at engraftment (e) and 30 days after engraftment (E30) for each patient for each oral site. The thick line indicates the median value at each timepoint. E75, 75 days after engraftment. \*\*$P$-value < 0.01; \*\*\*$P$-value < 0.001; and \*\*\*\*$P$-value < 0.0001.

diversity resistance, resilience, or stability (Fig. S11a). Compositional stability was also not correlated with diversity levels at P (Fig. S11b)

In addition, when representing samples from all timepoints using Principal Coordinate Analysis (PCoA), we noticed that confidence intervals for E samples were larger, indicating considerable inter-patient compositional variability under perturbation (Fig. S11c). To confirm this observation, we determined the most perturbed timepoint by

quantifying the extent of compositional shifts between timepoints. As presented in Fig. 6b, compositional changes were more pronounced between A and E. Next, we evaluated inter-patient compositional variability at each timepoint either by assessing the compositional distance between samples and the respective timepoint centroid (Fig. 6c) or by calculating for each timepoint all pairwise compositional distances (Fig. 6d). Both results confirmed maximum inter-patient compositional variability at E under maximized perturbation, underscoring that allo-HSCT modifies oral microbiota differently for each patient.

Finally, we investigated if this variability in oral microbiota dynamics during allo-HSCT influenced oral microbiota recovery after engraftment. Although our results indicate that post-engraftment samples overall occupy a similar compositional space in comparison to P, this does not necessarily imply that patients recover their respective initial oral microbiota compositions after engraftment. In order to evaluate oral microbiota compositional recovery per patient, we analyzed the compositional distance from P for each patient and each site during allo-HSCT and after engraftment. Interestingly, we noted that even though most patients showed a recovery trajectory after engraftment, some did not (Fig. 6e).

## Recovery of oral microbiota composition and allo-HSCT outcomes

To investigate whether oral microbiota recovery after allo-HSCT was associated with allo-HSCT outcomes, we grouped our patients based on their ability to recover their preconditioning composition. We calculated the compositional distance between P and E30 and classified patients as recoverers (distance < 0.5) or non-recoverers (distance ≥ 0.5). We further illustrate these contrasting recovery behaviors using PCoA with compositional trajectories of a representative OM recoverer and of an OM non-recoverer (Fig. 7a). PCoAs for each patient are presented in Fig. S12. Overall, 77%, 69%, and 77% of our patients recovered their initial GCF, OM, and SB microbiota composition after engraftment, respectively (Fig. 7b).

Next, we used univariate analysis to investigate whether oral microbiota recovery after allo-HSCT was associated with allo-HSCT outcomes (Table S3; Fig. S13). Interestingly, OM recovery was associated with prolonged overall survival [hazard ratio, HR (95% confidence interval, CI) = 0.17 (0.05–0.52), $P$-value = 0.002; Fig. 7c], prolonged progression-free survival [HR (95% CI) = 0.06 (0.01–0.34), $P$-value = 0.001; Fig. 7d], and a lower risk of underlying disease relapse [HR (95% CI) = 0.20 (0.06–0.69), $P$-value = 0.011; Fig. 7e]. OM recovery, however, was not associated with altered risk of transplant-related death, and GCF recovery or SB recovery was not associated with allo-HSCT outcomes (Table S3; Fig. S13).

To identify possible confounding variables, we used univariate analysis to investigate whether clinical parameters (including antibiotic usage; Table S1) were associated with allo-HSCT outcomes (Tables S4 to S7). We found that disease risk index (DRI), conditioning intensity, and DOT were significantly associated with OS (Table S4). DRI was also associated with PFS (Table S5) and the risk of underlying disease relapse (Table S6). We then used a multivariate analysis to assess whether OM recovery was an independent predictor of allo-HSCT outcomes (Table S8). OM recovery remained significantly associated with prolonged OS [HR (95% CI) = 0.09 (0.02–0.35), $P$-value < 0.001; Fig. 7f], prolonged PFS [HR (95% CI) = 0.09 (0.02–0.49), $P$-value = 0.005; Fig. 7g], and with a lower risk of underlying disease relapse [HR (95% CI) = 0.19 (0.06–0.55), $P$-value = 0.003; Fig. 7h].

## Underlying factors associated with oral mucosa microbiota recovery

Given the relevant associations between OM recovery and allo-HSCT outcomes, we searched for underlying factors associated with OM recovery. OM recovery was not associated with clinical parameters such as age, underlying disease, and graft source (Table S9). The usage of specific antibiotic classes, LOT, and DOT between P and E30 were

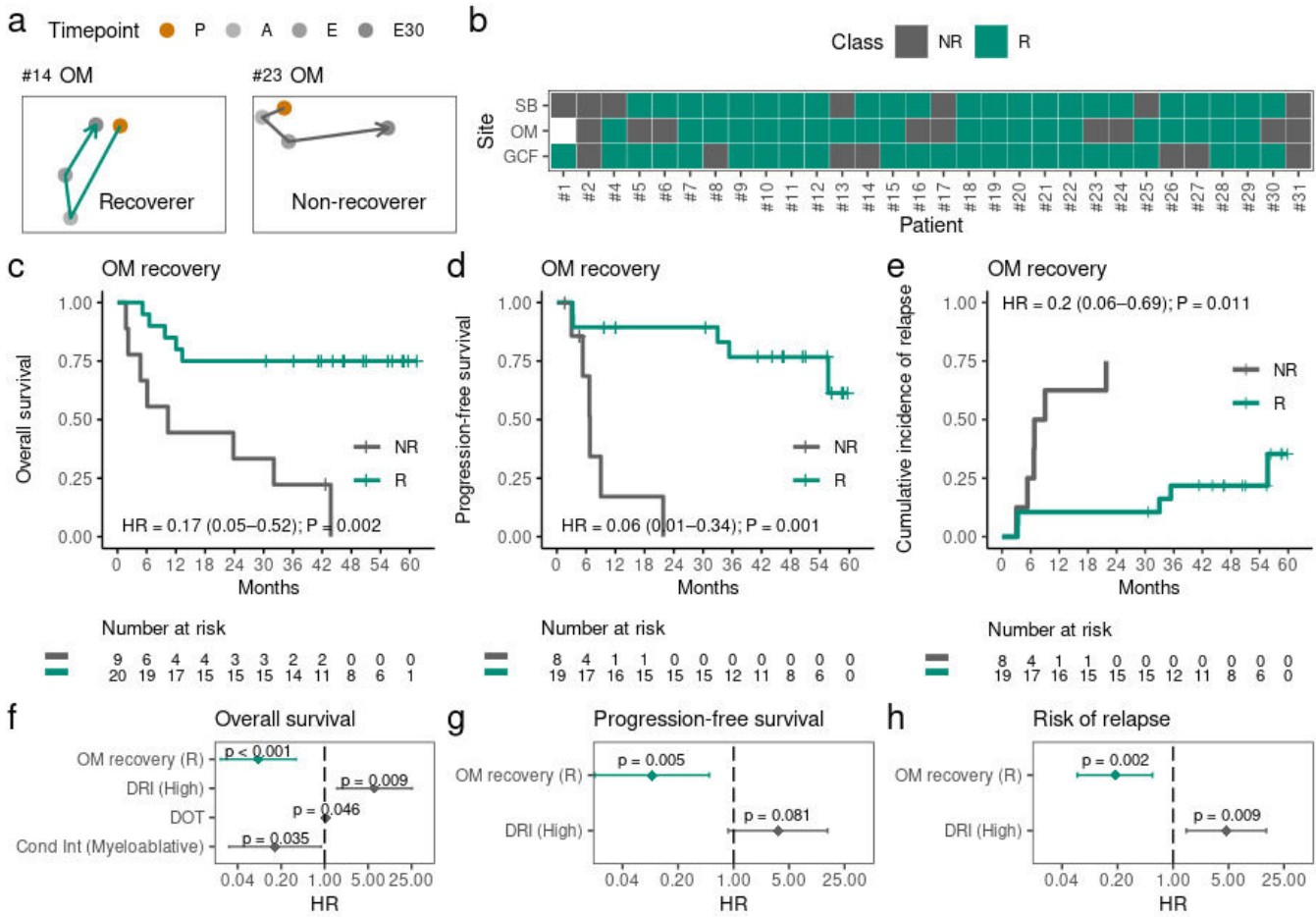

**FIG 7** (a) Principal Coordinate Analysis with representative microbiota trajectories of an OM recoverer and non-recoverer. (b) Recovery classifications per site for each patient. Patient #1 OM recovery could not be evaluated due to missing samples. (c and d) Kaplan-Meier curves comparing overall survival (c) and progression-free survival (d) among OM recoverers (R) and non-recoverers (NR). (e) Cumulative incidence curves of relapse among OM R and OM NR. (f–h) Multivariate analysis for overall survival (f), progression-free survival (g), and risk of relapse (h). Each model includes OM recovery and the clinical variables that are relevant for each outcome. E30, 30 days after engraftment and E75, 75 days after engraftment, HR, hazard ratio; DRI, disease risk index; DOT, days of antibiotic therapy; and Cond Int, conditioning intensity.

also not associated with OM recovery (Table S9; Fig. S14a). In addition, OM recoverers and non-recoverers showed similar intervals between stem-cell infusion and engraftment (Fig. S14b).

We also evaluated whether OM microbiota characteristics could be related to OM recovery. OM recoverers did not show higher OM diversity at E30 (Fig. 8a), indicating OM non-recoverers did not necessarily possess a dysbiotic OM microbiota at E30. In line with this, OM blooms throughout allo-HSCT were not more frequent among OM non-recoverers (Fisher's exact test, OR = 4.07, *P*-value = 0.13). On the other hand, OM recoverers showed higher OM diversity at P and E (Fig. 8a). In fact, there was a significant negative correlation between OM diversity at P and the compositional distance between P and E30 (Fig. 8b). This effect was not observed for GCF and SB (Fig. 8b).

Lastly, we investigated if the reconstitution of blood cell counts was associated with OM recovery (see File S3; Fig. 8c). Blood cell counts at P or E were not associated with OM recovery. Interestingly, however, OM recoverers showed higher leukocyte counts at E30, which is mostly due to significantly higher neutrophil and lymphocyte counts in this group. Furthermore, normal (within reference values) leukocyte counts at E30 were more frequently observed among OM recoverers compared to OM non-recoverers (16/20 vs 3/9, respectively; Fisher's exact test, *P*-value = 0.032), and OM recoverers presented

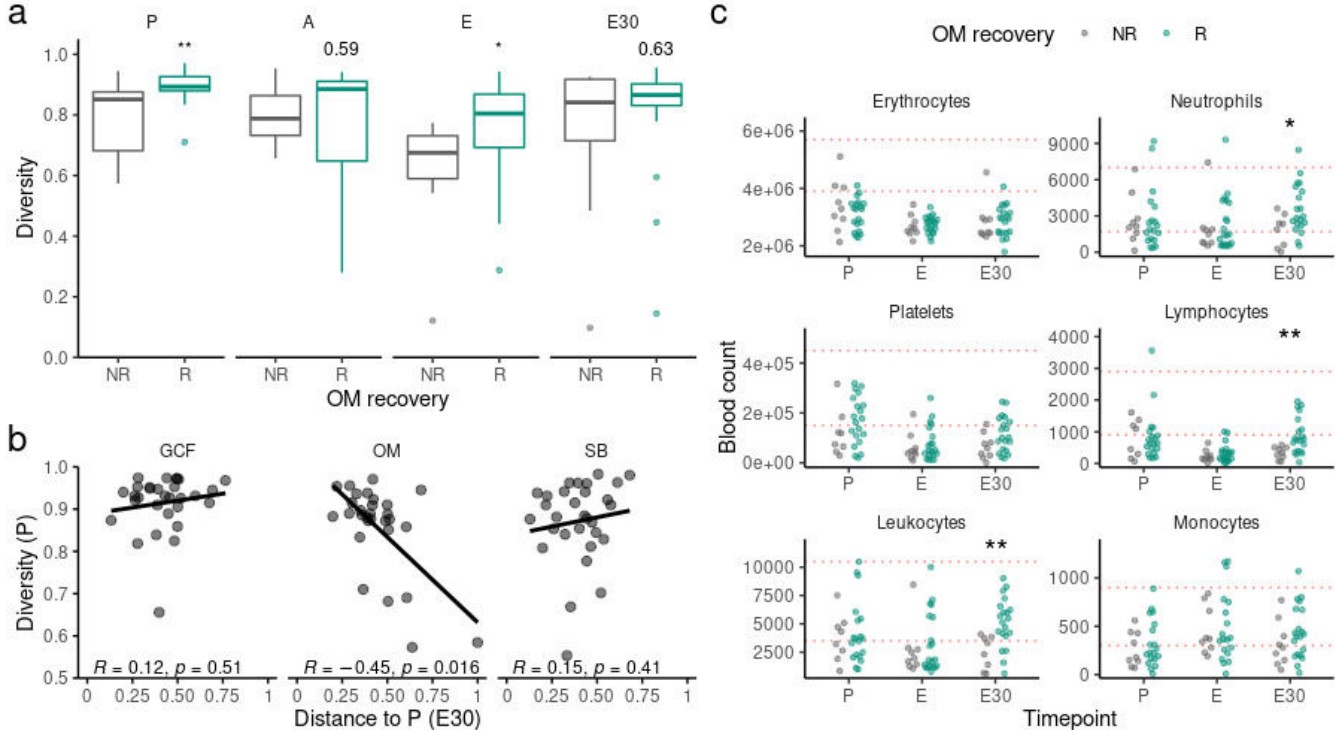

**FIG 8** (a) Diversity (Gini-Simpson) among OM recoverers and non-recoverers for each timepoint. Mann-Whitney *U* test was used. (b) Correlation between diversity (Gini-Simpson) at preconditioning (P) and the compositional distance (weighted UniFrac) between P and 30 days after engraftment (E30) for each oral site. Spearman's rank correlation test was used. (c) Blood cell counts among OM recoverers and non-recoverers per timepoint for each blood cell type. Red dotted horizontal lines indicate normal counts (within reference values). Mann-Whitney *U* test was used. E75, 75 days after engraftment. *P*-value < 0.05 and **P*-value < 0.01.

higher leukocyte counts throughout 1 year after allo-HSCT compared to non-recoverers due to the combined contribution of higher neutrophil, lymphocyte, and monocyte counts (Fig. S14c).

## DISCUSSION

The anatomical complexity of the oral cavity provides a multitude of physicochemical environments for microbes to thrive (1, 3). Although several dozen core bacterial genera inhabit all oral compartments, different species occupy each oral niche, meaning oral microbes are site specialists that compose distinct microbiota in each oral environment (1, 59). We and others have previously reported the impact of allo-HSCT in oral microbiota and their associations with allo-HSCT complications and outcomes (22–26, 30). However, these studies analyzed single oral sites and were mostly limited to the peri-engraftment period of allo-HSCT. To our knowledge, this is the first study to evaluate the impact of allo-HSCT in the microbiota of various oral sites simultaneously during and after allo-HSCT.

We found that the microbiota of all oral sites was severely damaged by allo-HSCT, but each site responded differently to the perturbations associated with allo-HSCT. Compositional differences between oral sites were lost during allo-HSCT and partially recovered after engraftment. Oral microbiota injury was marked by loss of diversity and emergence of opportunistic potentially pathogenic genera. Notably, these opportunistic genera could colonize all three oral sites and likely contributed to the loss of compositional differences between distinct oral microbiota observed after conditioning. Colonization by opportunistic genera was more common at E, explaining the higher compositional variability and lower diversity observed at E, which we found to be the most perturbed allo-HSCT phase for all oral sites. This is in line with the Anna Karenina

principle applied to host-associated microbiomes (60), which states that more diverse communities tend to be more compositionally similar, while perturbed communities tend to occupy several alternative dysbiotic states.

Blooms of opportunistic genera were associated with prolonged antibiotic exposure and the use of glycopeptides. This association is clinically relevant in the allo-HSCT setting since glycopeptide-resistant bacteria (e.g., vancomycin-resistant enterococci) are a common cause of infections in the hospital environment (61), especially in immuno-suppressed individuals. In addition, we observed that, in some cases, oral microbiota blooms preceded respiratory infections caused by the blooming bacteria, linking the oral microbiota dynamics during allo-HSCT to a common allo-HSCT complication (62), probably due to the translocation of oral bacteria to the respiratory tract through aspiration (56). Similarly to our study, Thänert et al. (63) showed pathobiont blooms in the gut microbiota often preceded urinary tract infections, but, as observed here, not all blooms were associated with subsequent infection (63). Interestingly, even though the mouth is a highly vascularized organ and the existence of an oral-blood translocation axis has been proposed (64), we did not find a clear association between oral bacteria blooms and bacteremia events during allo-HSCT.

Respiratory infections following blooms were caused by *E. faecium* in patient #1 and *S. maltophilia* in patient #2. *S. maltophilia* colonization has been reported in 7% of allo-HSCT recipients and is associated with higher non-relapse mortality risk due to higher odds of invasive *S. maltophilia* infections (65). Our results highlight that nosoco-mial bacteria such as *S. maltophilia* can colonize the oral cavity during allo-HSCT. These results point to the importance of maintaining oral health during allo-HSCT not only to prevent oral but also distal complications (e.g., hospital-acquired pneumonia) (56). Furthermore, our results suggest that tracking drastic oral microbiota changes during allo-HSCT may guide early interventions to prevent infections. This will be especially useful when the causative agent is not a common respiratory pathogen such as in the case of *E. faecium* (66).

Longitudinal analysis of oral microbiota diversity and composition showed the post-transplant oral microbiota was overall similar to the preconditioning microbiota, but patient-level analysis showed that 23%–31% of the patients did not recover their preconditioning microbiota composition. Variability in gut microbiota recovery following a perturbation has been previously described (67, 68), including after allo-HSCT, where most patients (>90%) were unable to recover their initial gut microbiota composition (68). The higher proportion of patients who recovered their preconditioning composition in our study suggests that the oral microbiota is more resilient to the perturbations associated with allo-HSCT than the gut microbiota. This result is in line with a previous study showing that the oral microbiota is more resilient than the gut microbiota to antibiotic perturbation (69).

Pre-perturbation microbiota characteristics, such as the presence of keystone bacteria, influence microbiota recovery (70). Here, although we did not find specific taxa directly contributing to microbiota recovery, we found that patients who recovered their OM microbiota composition after allo-HSCT showed higher preconditioning OM diversity, indicating that more diverse OM microbiota are more resilient to allo-HSCT. Our results converge on the insurance hypothesis, which proposes that high-diversity communities are less susceptible to perturbations (71). Interestingly, in our study, OM compositional recovery was neither associated with the use of specific antibiotics nor with the duration of antibiotic exposure. This is possibly because OM microbiota composition is less impacted by antibiotics, as evidenced by the lack of associations between antibiotic usage and OM compositional stability. Host genetics, reestablishment of normal diet, and reconstitution of the immune system are other possible drivers of microbiota recovery after allo-HSCT. Here, we showed that leukocyte blood counts at E30 were higher in patients who recovered their OM microbiota composition, indicating a close link between early immune system reconstitution and oral microbiota recovery. We can speculate that immune reconstitution allows stricter control of microbiota

compositions [e.g., via immunoglobulin A (72)], which, along with the reestablishment of microbial environment (e.g., normal diet), supports the recovery of the initial OM microbiota composition (73, 74).

The ability to recover the OM initial microbiota composition was associated with better allo-HSCT outcomes. However, it is unclear if OM microbiota recovery is just a consequence or also a driver of early immune reconstitution, thus having a causal role in the improved outcomes following allo-HSCT. Evidence from gut microbiota studies indicates that the latter hypothesis is plausible (75). For instance, recent studies have shown that specific gut microbes are associated with immune cell dynamics post-allo-HSCT (15, 76). Similarly, Miltiadous et al. (77) found that higher peri-engraftment gut microbiota diversity was associated with higher lymphocyte counts 100 days after transplant (77). In addition, murine model experiments showed that gut microbiota supports immune reconstitution by allowing a higher dietary energy uptake (78). Most importantly, in a controlled randomized clinical trial, patients who received autologous fecal microbiota transplant after allo-HSCT showed higher leukocyte counts 100 days after engraftment, indicating recovery of the gut microbiota has a causal role in facilitating immune system reconstitution (15). If this causal relationship extends to the oral microbiota, the use of therapeutic interventions to promote oral health and microbiota recovery in allo-HSCT recipients, such as oral microbiota transplants (79), could potentially improve allo-HSCT outcomes.

An important limitation of our study is its small sample size, which did not allow underlying disease stratification to parse the effect of different diseases on oral microbiota dynamics. Still, the longitudinal design, assessment of different oral sites, and evaluation of a Brazilian cohort [a population underrepresented in human micro-biome studies (80)] with extensive metadata publicly available are strengths of our study that should be highlighted. Also, to better address the influence of oral bacteria in immune cell dynamics, future studies will have to combine high temporal resolution oral microbiota data with more deeply phenotyped immune cell counts (e.g., flow cytometry data). In addition, since 16S rRNA amplicon sequencing has limited taxonomic resolution, further studies should ideally be performed using shotgun metagenomic sequencing, as this would allow strain-level dynamics tracking and functional potential evaluation. More powered and higher resolution data could point to potential mechanistic links for the associations described here. Finally, here and previously (24, 25), we showed that associations between gut microbiota and allo-HSCT outcomes broadly extend to the oral microbiota. However, studies with synchronous gut and oral microbiota profiling will be necessary to decipher how these microbiota are linked during allo-HSCT, especially considering the increased translocation of oral bacteria along the oral-gut axis during disease (81).

## Conclusion

We observed clear patterns of microbiota dysbiosis in all three oral sites during allo-HSCT; however, each oral site responded differently to the perturbations associated with allo-HSCT (e.g., antibiotic treatment). More importantly, oral microbiota injury and recovery patterns were associated with allo-HSCT complications and outcomes. This study shows how tracking oral microbiota injury and recovery in the allo-HSCT setting may improve our understanding of allo-HSCT clinical course and help deliver a safer and more effective treatment for allo-HSCT recipients.

## ACKNOWLEDGMENTS

V.H. was supported by Fundação de Amparo à Pesquisa do Estado de São Paulo (FAPESP, process no. 13996-0/2018). F.H.K. was supported by FAPESP (process no. 16854-4/2015).

E.R.F. and A.A.C. designed the study. V.C.M., L.T., V.R., Y.N., and C.A.-R. recruited and clinically evaluated the volunteers. V.C.M. collected data from clinical records. W.M.-S. collected oral samples. F.H.K. processed most of the samples. F.H.K. and P.F.A. performed

the sequencing. V.H. and A.A.C. conceptualized the analysis. V.H. performed all bioinformatics and statistical analyses. V.H., J.S.B., and A.A.C. contributed to the interpretation of the results. V.H. and A.A.C. wrote the original draft of the manuscript. V.H., J.S.B., V.C.M., P.F.A., C.A.-R., and A.A.C. reviewed and edited the manuscript. All authors read and approved the final manuscript.

## AUTHOR AFFILIATIONS

[1]Centro de Oncologia Molecular, Hospital Sírio-Libanês, São Paulo, Brazil

[2]Departamento de Bioquímica, Instituto de Química, Universidade de São Paulo, São Paulo, Brazil

[3]Hospital Nove de Julho, Rede DASA, São Paulo, Brazil

[4]Escola Paulista de Medicina, Universidade Federal de São Paulo, São Paulo, Brazil

[5]Centro de Oncologia, Hospital Sírio-Libanês, São Paulo, Brazil

[6]Hospital das Clínicas da Faculdade de Medicina, Universidade de São Paulo, São Paulo, Brazil

[7]Instituto do Câncer do Estado de São Paulo (ICESP), São Paulo, Brazil

## AUTHOR ORCIDs

Vitor Heidrich  http://orcid.org/0000-0001-6617-9187
Anamaria A. Camargo  http://orcid.org/0000-0002-6076-9597

## FUNDING

| Funder | Grant(s) | Author(s) |
|---|---|---|
| Fundação de Amparo à Pesquisa do Estado de São Paulo (FAPESP) | 13996-0/2018 | Vitor Heidrich |
| Fundação de Amparo à Pesquisa do Estado de São Paulo (FAPESP) | 16854-4/2015 | Franciele H. Knebel |

## DATA AVAILABILITY

The bioinformatics pipeline used to process the sequencing data, the R scripts used to run the analyses and generate the figures, and all clinical metadata (anonymized) necessary to reproduce these results are available at https://github.com/vitorheidrich/oral-microbiota-hsct. Raw sequencing data have been deposited in the European Nucleotide Archive (ENA) at EMBL-EBI under accession number PRJEB53914. Some samples (analyzed in past studies) were deposited previously in ENA at EMBL-EBI under accession numbers: PRJEB42862 and PRJEB49175.

## ETHICAL APPROVAL

This study was approved by the Ethics Committee of Hospital Sírio-Libanês (#HSL 2016–08), in line with the Declaration of Helsinki. All patients provided their written informed consent to participate.

## ADDITIONAL FILES

The following material is available online.

### Supplemental Material

**File S1 (Spectrum02910-23-s0001.pdf).** Timelines of antibiotic usage.
**File S2 (Spectrum02910-23-s0002.docx).** Tables S1 to S9 and Fig. S1 to S14.
**File S3 (Spectrum02910-23-s0003.docx).** Supplemental methods.

Open Peer Review

**PEER REVIEW HISTORY (review-history.pdf).** An accounting of the reviewer comments and feedback.

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
