## [Reviewer comments · Microbiology Spectrum]

Microbiology Spectrum

Longitudinal analysis at three oral sites links oral microbiota to clinical outcomes in allogeneic hematopoietic stem-cell transplant

Vitor Heidrich, Franciele Knebel, Julia Bruno, Vinícius de Molla, Wanessa Miranda-Silva, Paula Asprino, Luciana Tucunduva, Vanderson Rocha, Yana Novis, Eduardo Fregnani, Celso Arrais-Rodrigues, and Anamaria Camargo

Corresponding Author(s): Anamaria Camargo, Hospital Sírio-Libanês

Review Timeline:

Submission Date:	July 20, 2023
Editorial Decision:	September 15, 2023
Revision Received:	September 21, 2023
Accepted:	October 9, 2023

Editor: Zhenjiang Xu

Reviewer(s): Disclosure of reviewer identity is with reference to reviewer comments included in decision letter(s). The following individuals involved in review of your submission have agreed to reveal their identity: Na Fei (Reviewer #1); Irene Wagner-Dobler (Reviewer #2)

Transaction Report:

DOI: <https://doi.org/10.1128/spectrum.02910-23>

September 15, 2023

Dr. Anamaria A Camargo
Hospital Sírio-Libanês
São Paulo
Brazil

Re: Spectrum02910-23 (**Longitudinal analysis at three oral sites links oral microbiota to clinical outcomes in allogeneic hematopoietic stem-cell transplant**)

Dear Dr. Anamaria A Camargo:

Link Not Available

Sincerely,

Zhenjiang Xu

Journals Department
Reviewer comments:

Reviewer #1 (Comments for the Author):

Summary:

Heidrich et al. represent a study that reported an in-depth understanding of oral microbiota dynamics during and after allo-HSCT using 16S sequencing data. This study include 440 samples encompassing five timepoints and three oral sites: gingival crevicular fluid (GCF), oral mucosa (OM), and supragingival biofilm (SB), which allowed a longitudinal anatomically-aware analysis of the oral microbiota. Using 16S rRNA gene sequencing, the authors characterized the diversity, compositional, and taxonomical changes in oral microbiota during allo-HSCT and after engraftment. They associated these changes with antibiotic usage and allo-HSCT complications. Finally, they evaluated recovery trajectories after allo-HSCT to associate oral microbiota recovery with allo-HSCT outcomes and find that oral microbiota recovery is associated with allo-HSCT outcomes.

Strength:

- 1, This study is the first study to evaluate the impact of allo-HSCT in the microbiota of various oral sites ((gingival crevicular fluid, oral mucosa, and supragingival biofilm)) simultaneously during and after allo-HSCT including the longitudinal design in a Brazilian cohort of allo-HSCT recipients.
- 2, This study performed a thorough and in-depth analysis for the 16S rRNA gene sequencing data set, which add to the current useful bioinformatic knowledge.
- 3, This study showed that the oral microbiota injury and recovery patterns were highly informative on transplant complications and outcomes, which highlight the importance of tracking recipient's oral microbiotas changes during allogeneic hematopoietic stem-cell transplant to improve our understanding of its biology, safety, and efficacy.

Major comments:

- 1, Overall, the studies are associative in nature. More association or investigation is needed to conclude that the stability of the oral microbiota and its associations with allo-HSCT outcomes may offer a unique opportunity to identify predictive biomarkers and develop therapeutic interventions to promote oral health in allo-HSCT recipients, potentially improving allo-HSCT safety and efficacy.

For example, it is suggested to apply random Forrest machine learning to evaluate the prediction possibility of the recipient's oral microbiotas changes about allo-HSCT outcomes.

- 2, Many factors could have drastic influence on microbial structure. For example, diet is one of the main drivers for shifting the oral microbiota. Besides, the authors also concluded that unlike gut microbiota, oral microbiota composition is not impacted by antibiotics, as evidenced by the lack of associations between antibiotic usage and OM compositional stability. However, in the cohort, gender, diet or other clinical status were not considered, except antibiotics usage.

- 3, Line 370, "We next tested whether blooms at different oral sites were associated with antibiotic usage. E75 blooms were not considered in this analysis since our antibiotic usage survey focused on the period between P and E30 (see Materials and Methods)."

According to the data shown in the Figures 1 and 2, E75 is the timepoint for most of the microbiota recovered. It is necessary to include this time point to compare the blooms to other time points.

- 4, Line 407, "Taken together, these results suggest that the use of vancomycin during allo-HSCT allowed the emergence and the bloom of pathogenic *Stenotrophomonas melophilis* in oral microbiota, which later translocated to the respiratory tract, causing a respiratory infection."

This is an overstated conclusion without sufficient evidence to support.

- 5, We calculated the compositional distance between P and E30, and classified patients as recoverers (distance <0.5) or non-recoverers (distance {greater than or equal to}0.5).

Again, according to the data shown in the Figures 1 and 2, E75 is the timepoint E75 for most of the microbiota recovered. This time point needs to include to evaluate the recovery of microbiota.

- 6, Line 454, "Next, we used univariate analysis to investigate whether oral microbiota recovery after allo-HSCT was associated with allo-HSCT outcomes (Table S3; Fig. S13). Interestingly, OM recovery was associated with prolonged overall survival."

It is also possible that the oral microbiota recovery is resulting from the reduced antibiotic usage and it has not associated with allo-HSCT outcomes or even prolonged overall survival.

- 7, Line 506, "To our knowledge, this is the first study to evaluate the impact of allo-HSCT in the microbiota of various oral sites simultaneously during and after allo-HSCT."

Difference among 3 sites is not evaluated. Which oral site of the microbiota has the best association with allo-HSCT outcomes or even prolonged overall survival? What is the aim to evaluate the impact of allo-HSCT in the microbiota of various oral sites simultaneously? What does the results of the 3 sites' microbiota indicate, respectively?

- 7, Line 583: "If this causal relationship extends to the oral microbiota, the use of therapeutic interventions to promote oral health and microbiota recovery in allo-HSCT recipients, such as oral microbiota transplants (79), could potentially improve allo-HSCT outcomes."

In this study, the authors still failed to point out which bacterial lineages or function might contribute to this association between the oral microbiota and allo-HSCT outcomes, which will be important potential clinical therapeutic targets.

- 8, In the future studies, a plan or strategy to explore the underlying mechanism between the oral microbiota and allo-HSCT

outcomes need to be discussed.

Minor comments:

- 1, PCoA plot: add the variable of each PC in the plot.
- 2, line 276 recovery to baseline compositions (Fig. 1c). should be Fig. 2c
- 3, Line 280 state (Fig. 1d). should be Fig. 2d

Reviewer #2 (Comments for the Author):

The manuscript asks a highly relevant question, and accordingly its results might inform clinicians in the future. The paper is based on a substantial amount of clinical and microbiota data. The Abstract is clear and informative. The Introduction is a pleasure to read - neither too wordy, nor too short, emphasizing the important information and clearly pointing out the novelty of the current study. Methods are extensive, although I have to admit that I cannot judge some of the statistics - but it is clear that each and every statement is supported by statistical evidence. The Results then are logically reported and follow the figures. Increasingly refined questions are asked and then extracted from the data, and I particularly like the reporting of non-significant findings. The Discussion, again, is relatively brief and to the point. It acknowledges the weaknesses of the study, stresses the implications of the findings, and hints to future scientific questions. There are no overstatements, but a clear communication of the importance of the findings.

Additionally I would like to acknowledge the style of writing, which is a pleasure throughout. I have only a small grammatical comment. I think "microbiotas" is not possible. "Microbiota" is already a plural noun. One would state "different microbiota" or "diverse microbiota" or similar.

Staff Comments:

Preparing Revision Guidelines

Please return the manuscript within 60 days; if you cannot complete the modification within this time period, please contact me. If you do not wish to modify the manuscript and prefer to submit it to another journal, please notify me of your decision immediately so that the manuscript may be formally withdrawn from consideration by Microbiology Spectrum.

Longitudinal analysis at three oral sites links oral microbiota to clinical outcomes in allogeneic hematopoietic stem-cell transplant

Dear Editor and Reviewers,

We thank the Reviewers for their helpful suggestions and comments. Changes are marked in red in the revised version of the manuscript.

Reviewer #1

Summary:

Heidrich et al. represent a study that reported an in-depth understanding of oral microbiota dynamics during and after allo-HSCT using 16S sequencing data. This study include 440 samples encompassing five timepoints and three oral sites: gingival crevicular fluid (GCF), oral mucosa (OM), and supragingival biofilm (SB), which allowed a longitudinal anatomically-aware analysis of the oral microbiota. Using 16S rRNA gene sequencing, the authors characterized the diversity, compositional, and taxonomical changes in oral microbiota during allo-HSCT and after engraftment. They associated these changes with antibiotic usage and allo-HSCT complications. Finally, they evaluated recovery trajectories after allo-HSCT to associate oral microbiota recovery with allo-HSCT outcomes and find that oral microbiota recovery is associated with allo-HSCT outcomes.

Strength:

- 1, This study is the first study to evaluate the impact of allo-HSCT in the microbiota of various oral sites ((gingival crevicular fluid, oral mucosa, and supragingival biofilm)) simultaneously during and after allo-HSCT including the longitudinal design in a Brazilian cohort of allo-HSCT recipients.
- 2, This study performed a thorough and in-depth analysis for the 16S rRNA gene sequencing data set, which add to the current useful bioinformatic knowledge.
- 3, This study showed that the oral microbiota injury and recovery patterns were highly informative on transplant complications and outcomes, which highlight the importance of tracking recipient's oral microbiotas changes during allogeneic hematopoietic stem-cell transplant to improve our understanding of its biology, safety, and efficacy.

Response:

We thank the Reviewer for the careful evaluation of our manuscript and the overall positive comments about our work.

Major comments:

- 1, Overall, the studies are associative in nature. More association or investigation is needed to conclude that the stability of the oral microbiota and its associations with allo-HSCT outcomes may offer a unique opportunity to identify predictive biomarkers and develop therapeutic interventions to promote oral health in allo-HSCT recipients, potentially improving allo-HSCT safety and efficacy.

For example, it is suggested to apply random Forrest machine learning to evaluate the prediction possibility of the recipient's oral microbiotas changes about allo-HSCT outcomes.

Response:

We agree that our study is exploratory in nature and that further research is needed to translate our findings into clinical practice through intervention or the use of the oral microbiota as a biomarker. On the latter point, as suggested by the Reviewer, we evaluated whether it was possible to predict oral mucosa microbiota recovery from preconditioning oral mucosa microbiota compositions. We used random forest models on ASV-level relative abundances combined with 5-fold cross-validation (repeated 10x and averaged to calculate the mean area under the ROC curve). As shown below, with a mean AUC = 0.44 that does not outperform the null model (AUC = 0.50), we conclude it is not possible to predict recovery from preconditioning compositions. Although pre-conditioning diversity was associated with microbiota recovery, it is not possible to use pre-conditioning microbiota composition alone to predict recovery. Post-transplant microbiota composition should also be considered in order to capture the dynamics of microbiota recovery. Finally, we point out that we did not use a machine-learning approach in our manuscript due to our limited sample size and the lack of an independent validation cohort.

2, Many factors could have drastic influence on microbial structure. For example, diet is one of the main drivers for shifting the oral microbiota. Besides, the authors also concluded that unlike gut microbiota, oral microbiota composition is not impacted by antibiotics, as evidenced by the lack of associations between antibiotic usage and OM compositional stability. However, in the cohort, gender, diet or other clinical status were not considered, except antibiotics usage.

Response:

We agree that different host-associated and environmental factors shape the oral microbiota. Therefore, we indeed ensured that demographic (including gender) and relevant clinical parameters were not associated with oral mucosa microbiota recovery (section “Underlying factors associated with oral mucosa microbiota recovery”, line 474-479). We also checked whether those parameters were associated with allo-HSCT outcomes and built adjusted models considering the confounding variables (line 463-472). For instance, the association between progression-free survival and oral mucosa microbiota recovery was adjusted for the disease risk index (Table S5).

The nutritional regimen during hospitalization for allo-HSCT is prescribed at an institutional level and then rigorously controlled by nutritionists, according to the rules established by the Brazilian Health Regulatory Agency (Anvisa). After discharge, patients also follow strict dietary recommendations, which are important for instance to prevent food-associated infections. Therefore, because all patients had a very controlled and similar diet during the study, diet was not considered in our analysis.

Finally, as detailed in the section “Impact of antibiotic usage on oral microbiota dynamics” (line 366-377), we do not conclude antibiotics do not impact the oral microbiota composition, but that different oral sites are impacted at different levels, with the oral mucosa microbiota showing the weaker associations with antibiotics usage (Table S2). Then, we use this observation to give a possible explanation in the Discussion on why the associations between recovery and clinical outcomes were present only for the oral mucosa (line 560-564). To be more precise, we have rephrased line 562-564 as follows:

“This is possibly because OM microbiota composition is **not** impacted by antibiotics, as evidenced by the lack of associations between antibiotic usage and OM compositional stability.”

“This is possibly because OM microbiota composition is **less** impacted by antibiotics, as evidenced by the lack of associations between antibiotic usage and OM compositional stability.”

3, Line 370, “We next tested whether blooms at different oral sites were associated with antibiotic usage. E75 blooms were not considered in this analysis since our antibiotic usage survey focused on the period between P and E30 (see Materials and Methods).”

According to the data shown in the Figures 1 and 2, E75 is the timepoint for most of the microbiota recovered. It is necessary to include this time point to compare the blooms to other time points.

Response:

We decided not to include the E75 timepoint in the blooming analysis for two main reasons. First, because, as pointed out by the reviewer, at E75 the microbiota is most similar to the initial state. Therefore the number of blooms at E75 was, as expected, very limited (5 blooms in total, representing only 6% of the blooming events). Second, we were mainly interested in analysing the association between blooms and antibiotic usage, and, as stated in the original manuscript, information about antibiotic use was available only until 100 days after stem-cell infusion (line X). Even though E75 samples were collected at a median 74.5 days after engraftment, a

significant number of E75 samples were actually collected after 100 days of stem-cell infusion, and out of the window of antibiotic usage data collection.

4, Line 407, "Taken together, these results suggest that the use of vancomycin during allo-HSCT allowed the emergence and the bloom of pathogenic *Stenotrophomonas maltophilia* in oral microbiota, which later translocated to the respiratory tract, causing a respiratory infection." This is an overstated conclusion without sufficient evidence to support.

Response:

We agree. We have moderated the sentence as follows (line 406-409):

"Taken together, **it is possible to speculate** that the use of vancomycin during allo-HSCT **might have** allowed the emergence and bloom of pathogenic *Stenotrophomonas maltophilia* in oral microbiota, which later translocated to the respiratory tract, causing a respiratory infection."

5, We calculated the compositional distance between P and E30, and classified patients as recoverers (distance <0.5) or non-recoverers (distance {greater than or equal to}0.5). Again, according to the data shown in the Figures 1 and 2, E75 is the timepoint E75 for most of the microbiota recovered. This time point needs to include to evaluate the recovery of microbiota.

Response:

As previously mentioned we decided to exclude E75 from the recovery analysis due to the high variability in the dates of actual sample collection (from 60 to 131 days after engraftment). In addition, three patients relapsed and one died between E30 and E75, reducing the number of patients that can be included in the associations between recovery and clinical outcomes (since they experienced the event before sample collection). Finally, and most importantly, E30 is the first clinical follow-up after allo-HSCT, and, since our primary goal is to predict outcomes, the earlier we evaluate microbiota recovery the better.

6, Line 454, "Next, we used univariate analysis to investigate whether oral microbiota recovery after allo-HSCT was associated with allo-HSCT outcomes (Table S3; Fig. S13). Interestingly, OM recovery was associated with prolonged overall survival."

It is also possible that the oral microbiota recovery is resulting from the reduced antibiotic usage and it has not associated with allo-HSCT outcomes or even prolonged overall survival.

Response:

We agree with the reviewer that this is an intuitive and logical possibility since we initially showed that antibiotic usage (DOT) was associated with overall survival in the univariate analysis. However, we also showed that i) the association between oral microbiota recovery and overall survival remained significant after adjusting the analysis for antibiotic use (Figure 7f, line 467-470) and that ii) oral microbiota recovery was not associated with antibiotic usage (Figure S14a/Table S9, line 477-478). Therefore, it is reasonable to conclude that the association between microbiota recovery and overall survival is not due to differences in antibiotics usage among recoverers and non-recoverers.

7, Line 506, "To our knowledge, this is the first study to evaluate the impact of allo-HSCT in the microbiota of various oral sites simultaneously during and after allo-HSCT."

Difference among 3 sites is not evaluated.

Response:

Differences between the three oral sites were in fact evaluated. Please see the section "Compositional differences between distinct oral microbiota during allo-HSCT and after engraftment". We also compared how the microbiota at different oral sites responded to the perturbations associated with TCTH-alo (section "Oral microbiota dynamics during allo-HSCT and after engraftment", line 269-272) and showed that the gingival crevicular fluid microbiota has higher diversity resilience.

Which oral site of the microbiota has the best association with allo-HSCT outcomes or even prolonged overall survival?

Response:

We demonstrated that the oral mucosa is the only site with significant associations between microbiota recovery and allo-HSCT outcomes ().

What is the aim to evaluate the impact of allo-HSCT in the microbiota of various oral sites simultaneously? What does the results of the 3 sites' microbiota indicate, respectively?

Response:

We decided to evaluate different oral sites for two main reasons: i) each oral site houses distinct bacterial communities and, since this was an exploratory work, the evaluation of three oral sites allowed us to dive deep into the alterations caused by allo-HSCT in the microbial composition of the oral cavity ii) distinct oral microbiota could be associated with allo-HSCT in different ways. For instance, the gingival crevicular fluid contains species able to translocate through the gingival epithelium, reaching the circulation and causing bacteremia in allo-HSCT patients. Teeth-associated biofilms (such as supragingival biofilm) have high microbial biomass and can interact with host cells to modulate immune homeostasis. Finally, the oral mucosa represents the largest surface area of the mouth, and oral mucosa microbiota has a rapid turnover and is associated with treatment-related oral toxicities, such as oral mucositis.

Indeed, in the current and previous studies, we observed site-specific associations. For instance, in our previous work we showed that the supragingival biofilm is associated with graft-versus-host-disease (Heidrich et al., 2021, PMID: 34220852), while the oral mucosa microbiota is associated with cancer-related outcomes such as progression-free survival (De Molla et al., 2021, PMID: 34475459). We also observed associations between oral mucosa microbiota and treatment toxicity (Bruno et al., 2022, PMID: 36266464).

8, Line 583: "If this causal relationship extends to the oral microbiota, the use of therapeutic interventions to promote oral health and microbiota recovery in allo-HSCT recipients, such as oral microbiota transplants (79), could potentially improve allo-HSCT outcomes."

In this study, the authors still failed to point out which bacterial lineages or function might contribute to this association between the oral microbiota and allo-HSCT outcomes, which will be important potential clinical therapeutic targets.

Response:

We agree with the reviewer that identifying specific taxa is essential to determining a causal relationship between oral microbiota and all-HSCT outcomes. Indeed, associations between specific taxa and allo-HSCT outcomes and complications were described in previous studies from our group (Heidrich et al., 2021, PMID: 34220852; De Molla et al., 2021, PMID: 34475459; Bruno et al., 2022, PMID: 36266464). In the current study, because of the inclusion of the newly sequenced E30 and E75 post-engraftment samples, we focused on post-transplant microbiota recovery and the impact of microbiota recovery on clinical outcomes. Still, we did try to find associations regarding specific taxa with microbiota recovery. Unfortunately, we were unable to find consistent associations, probably due to the high complexity of the oral ecosystem. We now disclose these attempts in the following rephrased sentence of the Discussion section (line 555-556):

“Pre-perturbation microbiota characteristics, such as the presence of keystone bacteria, influence microbiota recovery (70). Here, **although we did not find specific taxa directly contributing to microbiota recovery**, we found that patients that recovered their OM microbiota composition after allo-HSCT showed higher preconditioning OM diversity, indicating that more diverse OM microbiotas are more resilient to allo-HSCT”

We also agree that finding microbial functions associated with poor outcomes could be even more relevant than finding associated microbial taxa. However, prediction of microbial functions from 16S amplicon data has many limitations (Heidrich & Beule, 2022, 10.1002/imt2.38; Djemiel et al., 2022, 10.1093/gigascience/giab090), so we decided not to use it in the present work. Instead, a more robust way of assessing the functional potential of the oral microbiota would be to perform shotgun metagenomic sequencing, which we plan to do in our future studies.

9, In the future studies, a plan or strategy to explore the underlying mechanism between the oral microbiota and allo-HSCT outcomes need to be discussed.

Response:

As mentioned above, we plan to perform shotgun metagenomic sequencing in future studies to better dissect the underlying mechanism between oral microbiota and allo-HSCT outcomes. This would allow evaluation of the microbial pathways and functions potentially associated with allo-HSCT outcomes, providing some mechanistic hints to explain the associations. Such hints could then be evaluated using animal models and in vitro studies. We added a new sentence in the Discussion section to mention this possibility (line 598-599):

“In addition, since 16S rRNA amplicon sequencing has limited taxonomic resolution, further studies should ideally be performed using shotgun metagenomic sequencing, as this would allow strain-level dynamics tracking **and functional potential evaluation. More powered and higher resolution data could point to potential mechanistic links for the associations described here.**”

Minor comments:

1, PCoA plot: add the variable of each PC in the plot.

Response:

All PCoAs in Fig. 1 have the same percentage of variance explained, so they share the same labels indicated in the bottom and right-side of Fig. 1A (x: Axis #1 [41.5%], y: Axis #2 [14%]). This is because all PCoAs were generated from the same ordination so the plots are comparable.

2, line 276 recovery to baseline compositions (Fig. 1c). should to Fig. 2c

Response:

Thank you. Fixed (line 276).

3, Line 280 state (Fig. 1d). should be Fig. 2d

Response:

Thank you. Fixed (line 280).

Reviewer #2

The manuscript asks a highly relevant question, and accordingly its results might inform clinicians in the future. The paper is based on a substantial amount of clinical and microbiota data. The Abstract is clear and informative. The Introduction is a pleasure to read - neither too wordy, nor too short, emphasizing the important information and clearly pointing out the novelty of the current study. Methods are extensive, although I have to admit that I cannot judge some of the statistics - but it is clear that each and every statement is supported by statistical evidence. The Results then are logically reported and follow the figures. Increasingly refined questions are asked and then extracted from the data, and I particularly like the reporting of non-significant findings. The Discussion, again, is relatively brief and to the point. It acknowledges the weaknesses of the study, stresses the implications of the findings, and hints to future scientific questions. There are no overstatements, but a clear communication of the importance of the findings.

Additionally I would like to acknowledge the style of writing, which is a pleasure throughout. I have only a small grammatical comment. I think "microbiotas" is not possible. "Microbiota" is already a plural noun. One would state "different microbiota" or "diverse microbiota" or similar.

Response:

We thank the Reviewer for the evaluation and appreciation of our work. Following the Reviewer's grammatical advice, we replaced all instances of "microbiotas" by "microbiota" or similar.

October 9, 2023

Dr. Anamaria A Camargo
Hospital Sírio-Libanês
São Paulo
Brazil

Re: Spectrum02910-23R1 (**Longitudinal analysis at three oral sites links oral microbiota to clinical outcomes in allogeneic hematopoietic stem-cell transplant**)

Dear Dr. Anamaria A Camargo:

Your manuscript has been accepted, and I am forwarding it to the ASM Journals Department for publication. You will be notified when your proofs are ready to be viewed.

Sincerely,

Zhenjiang Xu
Editor, Microbiology Spectrum
